# Prevalence of and Contributors to Food Insecurity among College Athletes: A Scoping Review

**DOI:** 10.3390/nu16091346

**Published:** 2024-04-29

**Authors:** Jamie Pacenta, Brooke E. Starkoff, Elizabeth K. Lenz, Amanda Shearer

**Affiliations:** 1The Ohio State University, School of Health and Rehabilitation Sciences, Columbus, OH 43210, USA; 2SUNY Brockport, Department of Kinesiology, Sport Studies, and Physical Education, Brockport, NY 14420, USA; egrimm@brockport.edu (E.K.L.);

**Keywords:** food insecurity, food security, collegiate athlete, student-athlete

## Abstract

Given the financial demands of attending college, the transition to new living situations, abrupt changes in social support, and overall lifestyle adjustments, college students are at an increased risk of food insecurity (FI) compared to the general population. Collegiate athletes experience an even greater risk of FI as a result of greater time commitments and energy demands associated with their sports. This heightened vulnerability poses a tremendous threat to student-athletes’ academic and athletic achievements. This study aims to address the prevalence and primary determinants of FI among collegiate athletes while providing potential solutions to navigate and alleviate the effects of diminished food security among this demographic. To address these aims, a total of 18 articles were selected from both peer-reviewed and gray literature. The U.S. Household Food Security Survey Module (US-HFSSM) survey tools were predominantly utilized across universities throughout the United States to gather data on FI. Student-athletes reported experiencing FI across various regions of the United States, including universities in the northeastern states *(n* = 5), the southwest region (*n* = 3), the southeast region (*n* = 3), the northwest (*n* = 1), and the Midwest (*n* = 1). Overall, FI prevalence rates ranged from 9.9% to 65%, and the most significant contributors included limited financial resources, time management, meal plans, and housing location/amenities. These findings highlight a need for screening, education, and interventions to address FI among collegiate athletes.

## 1. Introduction

Despite national concerns regarding food insecurity (FI), which affects 12.8% of the U.S. population, it remains a significantly underestimated issue among college students [1]. Food insecurity often goes undetected among college students, particularly amidst the demands of academia and the transition to independent living. Studies reveal that, on average, approximately 30% of college students experience FI during their academic careers, a rate more than double the national average [1,2,3,4,5,6,7]. Further research across diverse institutions identifies additional groups at greater risk of FI, including those living in urban communities, students of color, Pell Grant recipients, athletes without meal plans, first-generation students, and those with pre-existing FI [8,9].

Among the vulnerable population of college students are student-athletes, encompassing individuals engaged in intercollegiate athletic programs supported by their respective institutions. The Hope Center’s Real College Survey found that 24% of Division I athletes experience FI, with higher rates in Division II (26%), and two-year colleges (39%) [10]. Student-athletes face unique barriers to food access due to demanding academic and athletic schedules, specialized dietary needs, and regular travel away from home. Balancing demanding academic and athletic schedules poses a unique challenge for college athletes, often impacting their ability to access adequate nutrition to fuel their sports performance [8,11,12]. Although various factors influence food access for all college students, recognizing the distinct challenges faced by student-athletes is crucial for developing effective interventions tailored to this specific demographic.

Collegiate athletes are at even greater risk of FI particularly due to the increased energy demands associated with physical performance. The reduced caloric intake, specifically of nutrient-dense foods, associated with FI can impair athletic performance and potentially increase injury risk [13]. Moreover, FI-related decreases in energy and nutrient intake raise the risk of Relative Energy Deficiency in Sport (REDs), contributing to physical and psychological harm in athletes [14]. This vulnerability emphasizes the urgent need for research on FI in college athletes. Despite the growing awareness of FI among college students, this unique population with specific nutritional challenges has been largely overlooked.

Understanding the distinct barriers to adequate nutrition faced by college athletes compared to the general student population is essential for tailoring effective interventions. Although comprehensive studies exist for non-athlete students, a critical knowledge gap remains regarding FI among college athletes. This scoping review aims to shed light on the prevalence of FI in college athletes, explore the specific factors contributing to this issue within their unique context, and offer prospective solutions to mitigate FI in this population. Through this investigation, we sought to: (1) elucidate the prevalence of FI among college athletes from a variety of institutions; (2) identify the specific contextual factors that contribute to FI in this cohort; and (3) propose target solutions to mitigate FI and its negative consequences. Addressing these issues will not only directly benefit college athletes but also inform our broader understanding of FI within higher education.

## 2. Materials and Methods

### 2.1. Search Strategy

All procedures for the scoping review were conducted based on guidelines within the JBI Manual for Evidence Synthesis [15]. The protocol for the review was registered on 22 August 2023, and is publicly available on the Open Science Framework register (https://osf.io/3bhgx/?view_only=ab875b81858e4ed4b9d552b6a55b21a7 accessed on 26 April 2024). A total of four databases were used for the initial identification of primary research articles: (1) PubMed/Medline; (2) EBSCOHost; (3) SportDiscus; and (4) GoogleScholar. In each database, every combination of the following keywords was used: “food insecurity” OR “food security” OR “access to health* foods” OR “food supply” OR “healthy food availability” AND athlete* OR “athlete* college” OR “college athlete*”. To identify gray literature (unpublished reports of studies), theses/dissertations, abstracts, and newspaper articles were included in the database searches. To extend the comprehensiveness of searches, citations selected for full-text review were also used for a reference list search (backward reference search) and a cited reference search (forward reference search). All database searches, reference searches, and author communications were conducted between 10 August 2023 and 19 March 2024.

### 2.2. Study Eligibility Criteria

Eligibility criteria were constructed with the following inclusion criteria: (1) college students in the United States; (2) collegiate athletes (including NCAA and others); and (3) the examination of food insecurity. The title and abstract results from the systematic searches were imported into Covidence systematic review software (Veritas Health Innovation, Melbourne, Australia) (www.covidence.org) and were independently screened by two researchers. Those meeting the eligibility criteria underwent a full-text review, in which the full texts of all citations considered for further review were screened independently by two researchers based on the inclusion criteria. Any discrepancies during this process were considered by a third researcher acting as a tiebreaker. The records were reviewed and organized to represent individual study samples. Although eighteen records reported on studies that met the eligibility criteria, only eight primary, peer-reviewed, original research articles were included. An additional ten resources were included as gray literature, including theses, dissertations, and abstracts (Figure 1).

### 2.3. Data Extraction

Data were extracted from each study and independently checked for accuracy by one other researcher. Using a standardized form, the extracted characteristics included the authors, publication year, whether the source was gray literature or peer-reviewed, year/month of data collection, study design, study population, sampling and recruitment strategies, sample size, sample demographics, size and location of the university, FI measurement tool used, FI reference period, medium (online, in-person, etc.) of FI assessment, prevalence of FI, and contributors to FI.

## 3. Results

### 3.1. Study Characteristics

The basic characteristics of the eighteen studies, which met all eligibility criteria, are outlined in Table 1. Less than half of the studies were considered primary articles, whereas the majority (56%) of the included studies were gray literature, including academic theses and dissertations. All publications included in this review were cross-sectional, and the sample sizes ranged from 10 to 3506 participants, with a median of 88 participants per study. To understand the prevalence of and the main contributors to FI, the results included institution types, participation demographics, and the tools used to collect and assess FI.

### 3.2. Institution Types

Overall, the majority of the included studies were conducted at National Collegiate Athletics Association (NCAA) institutions (*n* = 16), with one reporting on Army Reserve Officers’ Training Corps (ROTC) cadets and another with no description of the type of university. Of the sixteen studies including NCAA institutions, seven focused on DI, two focused on DII, three were completed at DIII institutions, and four included multiple divisions or did not note a specific division. Furthermore, nine studies occurred at public universities, two at private institutions, and three reported on students from multiple institutions. Four of the reported studies did not include information about the type of institutions where the research was conducted. The locations of the institutions were throughout the United States, including the northeastern states (*n* = 5), the southwest region (*n* = 3), the southeastern region (*n* = 3), the northwest (*n* = 1), and the Midwest (*n* = 1), with one study not reporting the location and four reporting on multiple institutions.

### 3.3. Participant Demographics

The demographic information included race/ethnicity, sex, sexual orientation, housing situation, income, meal plan access, and first-generation student status. Although three studies reported on all males, one reported on all females, and three did not include information regarding sex, 47% of the participants self-reported as female. Among the twelve studies that reported race and ethnicity, the median was 77.3% white students. One study reported the sexual orientation of their students, which included LGBTQ+ individuals. Of the five studies examining housing, the majority of participants lived on campus, with a median of 80% living on campus. Three studies assessed the relationship between FI and first-generation students, with Bowman et al. (2020) [23] reporting solely on first-generation college student-athletes. Additionally, three studies examined meal plan status and found, on average, that 86.2% had a meal plan.

### 3.4. Assessment Tools for FI

The USDA screens individuals for FI by analyzing results from food security survey modules [27]. The 18-item U.S. Household Food Security Survey Module (US-HFSSM) is a three-stage survey that allows for minimal respondent burden with the benefit of reliable data. In addition to this screening tool, a 10-item U.S. Adult Food Security Survey Module and a 6-item Short Form of the Food Security Survey Module provide a more condensed version of the survey. Goldrick-Rab et al. (2020) [10] utilized the 18-item survey, four studies employed the 10-item survey, and eight studies used the 6-item survey. In addition to the US-HFSSM, Brown et al. (2023) [8] utilized a series of researcher-created questions for students to answer regarding FI, whereas Hickey et al. (2019) [13] utilized a hunger survey developed specifically for assessing food security. Poll et al. (2020) [17] used a questionnaire on the childhood history of FI to gather further information on food security status.

### 3.5. Prevalence

Apart from one study highlighting only students identifying as food insecure, one describing the prevalence based on the type of institution, and another not reporting FI prevalence, fifteen reported a range of FI from 9.9% to 63%, with a median of 39.6% of participants identifying as food insecure. The lowest prevalence rate, reported at 9.9%, was observed among male-only participants attending the University of Mississippi, an NCAA DI institution, and the highest rate, 63%, was also observed among male-only participants attending an NCAA DI institution located in the southeastern United States. Furthermore, Brown et al. (2023) [8] identified trends in FI, with the greatest rates among participants identifying as Native Hawaiian or Pacific Islander (100%), without a meal plan (29.9%), receiving a Pell Grant (26.5%), first-generation (27.2%), and having a history of FI before college (52.5%). Lastly, Goldrick-Rab et al. (2020) [10] found the prevalence of FI to be greater among those attending two-year institutions (39%) compared to DI (24%), DII (26%), and DIII (21%) institutions.

### 3.6. Contributors

The most significant contributors to FI included limited financial resources, time management, and housing location/amenities. Overall, 50% of the studies reported limited financial resources as their primary cause for FI. Time management was another major contributor, with 44.4% of studies reporting that athletic commitments disrupted mealtimes and the ability to access campus dining resources. Lastly, 38.8% of the included studies reported the location of dining facilities and/or access to kitchen amenities and 22.2% reported the lack of options for specific dietary needs as the most substantial hurdle to food security.

Other less commonly reported contributing factors to FI included race/ethnicity (11.1%), sex (11.1%), age (5.6%), sport (5.6%), a history of FI (11.1%), social/access/personal factors (5.6%), location (5.6%), meal plans (11.1%), assistance from coaches and the institution (5.6%), being first-generation (5.6%), identifying as a Pell Grant recipient (5.6%), NCAA policy changes in the regulation of feeding among DIII institutions (5.6%), and the change in campus routines during and following COVID-19 (11.1%).

## 4. Discussion

This review aims to shed light on the prevalence of FI among college athletes, to explore the specific factors contributing to this issue within their unique context, and to offer prospective solutions to alleviate and prevent further FI among this population. The overall prevalence rates of FI ranged from 9.9% to 65%. Food insecurity rates were primarily captured using the US-HFSSM 18-, 10-, and 6-item surveys. The results indicate that the most reported contributors to FI among athletes included limited financial resources (50%), limited time (44.4%), the location of eateries (38.8%), and a lack of options for their dietary needs (22.2%). The evidence of FI among these universities highlights the need for immediate intervention.

### 4.1. Financial Challenges

The demands of playing a college sport and managing finances are challenging tasks for student-athletes across all collegiate levels and types of universities. One study examining 91 female student-athletes found that 25% of participants reported limited finances as the primary barrier to adequate food intake [20]. Similar studies have reported limited financial resources as the major contributor to FI for student-athletes [4,7,10,11,19,20,24]. Student-athletes often face additional expenses, such as equipment and travel fees, that the general college population is not expected to pay, which may partly explain these financial challenges [10]. To further compound the financial strain, the considerable time commitment to sports and academics often prevents collegiate athletes from working to earn more money.

The challenges of juggling an athletic schedule, coursework, and employment often force student-athletes to prioritize spending. As a result of financial prioritization, student-athletes may resort to eating less, skipping meals, or eating more affordable but less nutritious meals [7,10,11,19,24]. Based on the results from a food security questionnaire, 72% of participants stated that they often or sometimes worried about food running out before obtaining enough money to buy more, 73% stated that they often or sometimes felt that the food they bought did not last because there was not enough money to get more, and 38% stated that they were hungry but avoided eating because there was not enough money to buy more food [25].

These results demonstrate how the overall intake of nutritious and balanced meals among student-athletes is often sacrificed to save money. A survey conducted in the California University system found that food-insecure students were more likely to purchase food based on cost and not nutritional quality when compared to food-secure students [28]. An additional study found that 55% of polled student-athletes were not able to afford balanced meals due to their limited financial resources [25]. Even with the help of scholarships and supplemental food assistance program participation, these factors, which have been linked to assisting lower-income individuals, are not protective enough to prevent FI among students [20]. Although the prevalence of student-athletes receiving aid has not been investigated, Goldrick-Rab et al. (2020) [10] found that 18% of college students received food assistance benefits despite the significant list of eligibility criteria, which include working at least 20 h per week, being a single parent, or participating in on-the-job training [10,29].

### 4.2. Meal Plans

Meal plans are often not effective enough on their own to prevent FI among college athletes [30]. A study that looked at FI among four-year colleges reported a staggering prevalence of 43%, even though students had access to a meal plan and campus dining locations [30]. Out of the students enrolled in the meal plan option, those reporting higher rates of FI reported consuming fewer meals in the dining hall compared to their peers, with 69% of participants eating nine or fewer meals in a dining hall each week [30]. The same report also found that individuals consuming five to nine meals per week at a dining hall reported 7% higher rates of FI compared to individuals who consumed fewer than five meals per week, which shows that the number of meals per week and meal plans alone are not predictive of FI [30]. Lower rates of meals consumed per week while on a meal plan may decrease because off-campus students use swipes on top of purchasing groceries and cooking at home. One of the main concerns is that although some students can utilize on-campus dining locations, not all universities provide affordable meal plans [20,24]. Historically, research on college meal plans and dining options has focused on college students. As a result of limited research, similar rates among college athletes can be assumed.

### 4.3. Time

Although every student is responsible for time management, student-athletes face the additional challenge of managing their rigorous academic and athletic schedules and obligations, which has been shown to increase the rates of FI. Due to their commitment to athletics (e.g., practice, travel, and competitions), college athletes have less time for employment opportunities and mealtimes [4]. With these factors to consider, athletes must prioritize their mealtimes, which can be challenging, especially when many dining halls have limited hours of operation. Competition and practice times were often reported as interfering with dining hall hours, preventing athletes from accessing healthy meals on campus during appropriate mealtimes [4,7,8,11,19,20,25].

Many students struggle to find enough time in their schedules to prioritize grocery shopping, cooking, and eating because of their busy academic and athletic schedules [11]. Even though many athletes have a meal plan, they often report being too busy or too tired to cook for themselves [11]. Among a cohort of 787 student-athletes, 45.4% reported that practice hours interfered with dining hours and 22% reported that game times interfered with dining hours [8]. Student-athletes must obtain the appropriate number of calories to fuel their daily needs, including a demanding physical component. Even though athletes may understand their need to eat for sport, achieving those needs may be especially challenging due to their demanding schedules and the limited mealtimes provided by the institution.

### 4.4. Housing

The living environment of student-athletes also plays a significant role in the risk of FI. For those living off campus, the cost of rent in addition to other living expenses may increase the risk of FI. Living off campus with limited finances was reported as a major barrier to adequate food access for student-athletes [7]. Additionally, the increase in regional off-campus housing costs over the last few years may be related to the increased prevalence of FI among student-athletes [7]. Similarly, on-campus housing comes with an additional set of challenges. Dormitories and on-campus living quarters offer varying types of living arrangements and amenities which can influence students’ food security status. Students living on campus at a public university in New England reported limited access to a kitchen as one of the many contributing factors to FI, as they must go out of their way to access resources associated with cooking and preparing meals [4,11]. Although some of the dorms offer amenities specific to cooking and preparing meals, this resource is not always guaranteed and is unique to each university.

Surrounding campus resources, such as grocery stores and eateries, should be considered when observing FI rates among student-athlete populations. A study assessing campus locations and the rates of FI found that the incidence of very low food security was two times greater in an urban compared to a rural setting [9]. Access to healthy food is also significantly impacted by the neighborhood and the surrounding environment. Food deserts are often located where there are smaller populations, elevated rates of abandonment or deserted homes, and residents with limited education, lower income, and higher levels of unemployment [31].

### 4.5. COVID-19

A lack of access to nutritious and readily available food was heightened as students navigated life during the COVID-19 pandemic. Among 2,018 college students, 15% were newly food insecure as a direct result of the pandemic [32]. The drastic lifestyle changes that occurred for Americans during the pandemic also contributed to major shifts in the day-to-day lives of college-athletes, potentially increasing the risk of FI. COVID-19 safety precautions led to the closure of athlete fueling stations, reducing access to free and healthy snacks [7]. Additionally, campus closures during the pandemic directly resulted in a decline in food production and access to healthy foods, which placed those already experiencing FI in a more vulnerable position [33]. In response to campus closures, students returned home, which may have negatively or positively impacted their access to food, depending on the existing factors at home. Although some students experienced lower rates of FI when moving back home, this was not the same situation for those living independently. For students living independently, FI rates increased along with stress levels, poor health status, and the number of hours worked [34]. The pandemic exacerbated the challenges that college student-athletes were already facing while navigating living independently and brought a level of awareness to the public regarding FI among this population. Post-pandemic findings identified the widespread impact of FI on students’ overall health, wellness, and academic and athletic performance [7,8,11,19,20,25]. Given the connection between the pandemic and rates of FI, future studies may demonstrate the lasting impacts that the pandemic had on student-athletes’ access to adequate nutrition.

### 4.6. Impact of FI among Student-Athletes

The impact of FI on student-athletes has not been thoroughly examined, yet previous research highlighting FI among college students in general provides some information. Approximately 66% of student-athletes agreed that access to food and snacks would increase overall academic and athletic performance [20]. A specific concern for student-athletes is REDs, a common issue among athletes who do not consume enough calories. Low energy consumption can contribute to hormonal and menstrual alterations, reduced physical performance, decreased concentration and coordination, depression, mood alterations, and injury [7]. Additionally, many student-athletes do not receive nutrition education from a qualified practitioner, such as a registered dietitian nutritionist (RDN), to help guide the types of foods that are appropriate for supporting and even enhancing their performance goals [35]. Subsequently, if athletes fail to meet their energy needs, they risk suboptimal performance and injury, which may result in less playing time and the potential loss of scholarship.

### 4.7. NCAA Feeding Regulations

It is important to note that the resources available to student-athletes differ based on NCAA division. The NCAA significantly influences food access for college athletes, particularly at NCAA institutions. In 2014, the NCAA approved a rule allowing unlimited meals and snacks for DI student-athletes, specifically addressing FI. Prior to this change, student-athletes received either three meals a day or a food stipend. However, this policy only benefits DI athletes, excluding over 122,000 and 195,000 students at DII and DIII institutions, respectively, from improved food security. Notably, DII institutions have the highest rate of first-generation students (20%) compared to DI and DIII, exacerbating FI for those already at risk [36,37,38]. In 2020, the NCAA introduced a new policy for DII and DIII athletes, allowing institutions to provide snacks and permissible nutritional supplements when participating in competition (bylaw 16.5.1. (e)) [39].

Additionally, the recent Name, Image, and Likeness (NIL) policy enables college athletes to secure endorsements, potentially reducing FI rates. Although NIL provides financial opportunities for athletes, it may also exacerbate disparities. Athletes in states with limited or no NIL guidance may experience an uneven distribution of resources. To comprehensively address FI, the NCAA must continue monitoring and adapting policies to support student-athletes’ well-being.

### 4.8. Intervention Strategies

To better support student-athletes and diminish the prevalence of FI, campus interventions have been deemed an appropriate place to start. One basic primary step that team coaches can take is to screen for FI before athletes even arrive on campus. Coaches and staff can also use this screening as an opportunity to discuss the contributors to and risks of FI and provide resources to aid those in need. Some campuses have introduced screening programs to detect FI to proactively address the situation, providing access to support staff who help with time management, thus providing options on where to eat and how to plan meals in response to their busy schedules [4].

In response to the growing concern about FI on college campuses, The College and University Food Bank Alliance created a Student Government Toolkit Guide to provide directions for running a campus-led food pantry. This resource provides directions on how to allocate needs surveys to students, advice on partnering with regional food banks, setting up a pantry, and tips on operating a pantry [40]. Although there is no simple solution to solving FI on college campuses, providing accessible student-focused interventions, such as the Student Government Toolkit Guide, which is a valuable resource to utilize when determining intervention strategies.

## 5. Conclusions

The overall findings from this scoping review demonstrate the significant prevalence of FI among college athletes. Among those experiencing FI, the primary contributors include limited financial resources, overwhelming time commitments, the location of resources, and housing arrangements. Lesser reported contributors include limited kitchen access and cooking skills, increased energy needs, a family history of FI, disordered eating, a lack of support from family members or the university, and the COVID-19 pandemic. Research in this realm is crucial, especially when advocating for policy change surrounding the needs of our collegiate athletes. Although athletes continue their college journey, there is a need for up-to-date solutions to prevent the climbing rates of FI among athletes. Athletic programs are advised to implement screening tools and assessments to gauge athletes’ food security status and provide resources, such as fuel stations, created specifically for athletes.

## Figures and Tables

**Figure 1 nutrients-16-01346-f001:**
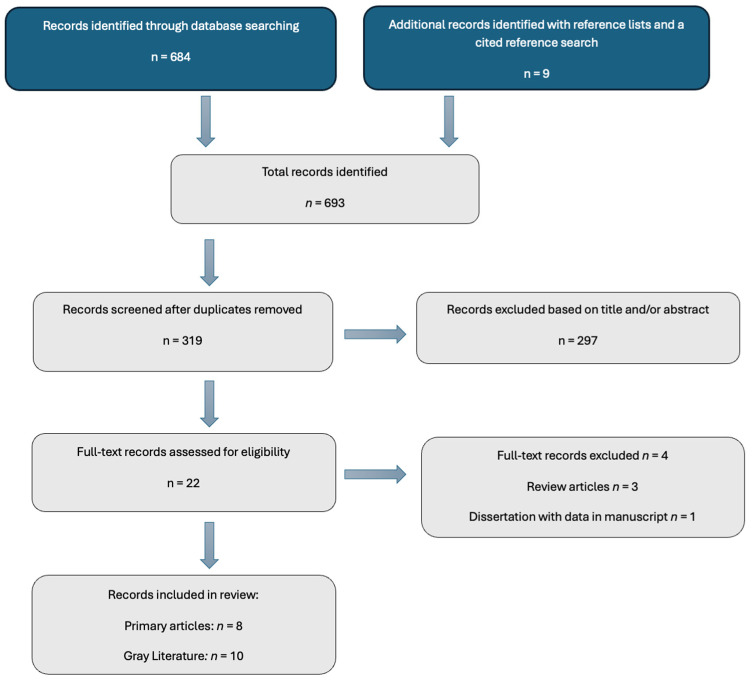
Study selection flow diagram.

**Table 1 nutrients-16-01346-t001:** Description of the included studies (*n* = 18).

Citation	Type of Institution	Participant Demographics	Tool to Assess FI	Prevalence of FI	Contributors to FI
Peer-Reviewed Articles					
Anziano & Zigmont, 2023 [11]	Public university in New England	-NCAA athletes (division not noted) -*N* = 10 -Food insecure -White: 90% -Females: 50% -On campus: 80.0%	6-item US-HFSSM	100%: only surveyed those with FI	-Lack of time -Special dietary needs -Limited campus dining options -Lack of healthy options in dining hall -Limited kitchen access -Limited access to transportation
Brown et al., 2023 [8]	-Multiple institutions -Unspecified location and type	-NCAA DIII -*N* = 787 -Female: 63.3% -White: 81.5% -First generation: 19% -Pell recipient: 18.2% -Live on campus: 81% -Have a meal plan: 83.3% Family Income: -<$25,000: 5.4% -$25,000–49,999: 6.5% -$50,000–74,999: 16.5% -$75,000–99,999: 12.9% -$100,000+: 39.8%	5 questions from 6-item US-HFSSM and 17 researcher-created questions	Overall: 14.7% By ethnicity: -White: 13.3% -Hispanic: 18.3% -Black: 31% -Asian: 8.5% -NHPI: 100% By meal plan: -With: 11.5% -Without: 29.9% By Pell Grant: -Yes: 26.5% -No: 11.1% First Generation: -Yes: 27.2% -No: 11.3% FI before college: -Yes: 52.5% -No: 11.5%	-Games during dining hours -Living off campus and/or limited money -Practice during dining hours -Regulation and restriction of feeding in DIII
Daniels & Hanson, 2021 [16]	Public land-grant research university in Kansas	-Army ROTC cadets -*N* = 37 -Female: 30% -White: 86.5%	6-item US-HFSSM	27%	-Social -Access -Personal
Douglas et al., 2022 [4]	Public university in rural East Texas	-NCAA DI -*N* = 78 -Female -White: 75.6%	6-item US-HFSSM	32%	-Timing of practice -Limited dining hall hours -Lack of financial resources -Lack of cooking skills and equipment
Goldrick-Rab et al., 2020 [10]	171 2-year and 56 4-year institutions across the U.S.	-13 NCAA DI -11 NCAA DII -24 NCAA DIII -124 2-Year Colleges -*N* = 3506	18-item US-HFSSM	-DI: 24% -DII: 26% -DIII: 21% -2-Year institutions: 39%	Limited financial resources
Hickey et al., 2019 [13]	Public liberal arts university in New Hampshire	-NCAA DIII -*N* = 371 (not all athletes) -Female: 65.8% -Athletes: 78.17% -White: 89.8% -Have a meal plan: 80.8% -First generation: 24.9%	Survey developed specifically for the study	34.6%	None reported
Poll et al., 2020 [17]	Public research university in Mississippi	-NCAA DI -*N* = 111 -Male	Childhood History of Food Insecurity Questionnaire	9.9%	FI before college
Reader et al., 2022 [7]	State University in Northwest U.S.	-NCAA DI -*N* = 45 -Female: 73.33% -White: 68.89% -On campus: 44.4%	10-item US-HFSSM	60%	-Balancing academics and athletics -Elevated energy needs -COVID-19 -Living location -Lack of financial resources
Abstracts					
Chimera et al., 2022 [9]	Public university in rural North Carolina and public research university in urban Alabama	-NCAA DI -None reported	10-item US-HFSSM	50%	Greater in urban vs. rural
Dellana et al., 2023 [18]	Public university in rural North Carolina and public research university in urban Alabama	-NCAA DI -*N* = 404 -LGBTQ+: *N* = 24	10-item US-HFSSM	45.6%	None reported
Gagnon et al., 2023 [19]	Not reported	-*N* = 124 -Female: 55% -White: 66%	Researcher developed survey	65%	-Financial insecurity -Dining hall hours -COVID isolation
Mayeux et al., 2020 [20]	Public university in rural East Texas	-NCAA (no division noted) -*N* = 91 -Female: 85.7% -White: 67%	6-item US-HFSSM	39.6%	-Lack of financial resources -Lack of time
Poll et al., 2017 [21]	University in southeast	-NCAA DI -*N* = 93 -Male -White: 48.4%	6-item US-HFSSM	16%	None reported
Theses/Dissertations					
Anziano, 2020 [22]	Public university in Connecticut	-NCAA DII -*N* = 18 -White: 88.9% -Live on campus: 83.3% -Female: 50% Hours worked per week: -0: 66.7% -1–12: 22.2% -12+: 11.1% Financing college: -Self-pay: 27.8% -Scholarships/grants: 55.6% -Loans: 38.9% -Assistance from others: 50% Meal plan: -None: 5.6% -Unlimited: 61.1% -Declining balance: 33.3%	6-item US-HFSSM	44.4%	-Lack of time -Family history -Spending priorities -Transportation -Limitations of dining halls -Meal plan -Limited kitchen access -Lack of assistance from coaches/universities
Bowman, 2020 [23]	Private Catholic university in Pennsylvania	-NCAA DII -*N* = 31 -First generation -Male: 71% -White: 55%	10-item US-HFSSM	40%	-Older students -Male -Female
Misener, 2020 [24]	Private liberal arts college in northeast	-NCAA DIII -*N* = 424 -Female: 46.5% -White: 79%	6-item US-HFSSM	31.8% in season	-Greater in male vs. female -Greater in white vs. non-white -Based on sport -Ran out of money for swipes -Ran out of money for campus food court -Unable to afford balanced meals -Correlated with receiving grant money -Correlated with being first-generation
Nilsson, 2023 [25]	University in southwest	-NCAA (division not noted -*N* = 70 -Female: 56.25% -Living location:-Campus housing: 28.13% -Off-campus, walking distance: 26.56% -Off-campus, driving distance: 45.31%	10-item US-HFSSM	Not reported	-Dining hall hours conflict with practice/game times -Living location -Limited resources (money)
Stowers et al., 2022 [26]	University in southeast	-NCAA DI -Football players -*N* = 85 -Male	10-item US-HFSSM	63%	Greater in black vs. white

Note: FI = food insecurity; NCAA = National Collegiate Athletics Association; 6-/10-/18-item US-HFSSM = U.S. Household Food Security Survey Module: 6-/10-/18-Item Short Form [16]; NHPI = Native Hawaiian/Pacific Islander.

## Data Availability

The original contributions presented in this study are included in the article, further inquiries can be directed to the corresponding author.

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
