# Peer review of "Prevalence of and Contributors to Food Insecurity among College Athletes: A Scoping Review"

_nutrients, 2024, doi:10.3390/nu16091346_

Round 1
Reviewer 1 Report
Comments and Suggestions for Authors
General Comments
Overall, I found this to be an interesting idea and research topic. In the introduction, the authors provide good rationale for why this topic is important. There may be a few things that the authors probably should address when conceptualizing the paper. One factor is that of scholarship status, not all sports receive the same scholarship. Sports such as football and basketball, unless walk-ons, are given a full scholarship which includes meal plan, but for many sports they may be receiving partial scholarships and meal plans may or may not be involved or different sports might receive different meal plans. Also, the NCAA has changed the ability of university’s to give food and drink to athletes (often called fueling stations). It might be interesting to discuss this as well. Both might not be able to be addressed in the research, but probably warrant some discussion in the introduction. Below are a few other minor things the authors should consider as well.
Specific comments
Lines 27-29; does this article address college athletes or is there another article the authors can cite regarding athletes?
Lines 74-75; it might be nice if the authors embedded the link to the Open Science Framework register in the text.
Line 76; is there a reason why SportDiscus was not used?
Line 78; is there a reason why college was not also used? Or maybe even higher education (not sure if this one would add)
Lines 89-90; can the authors talk about what happened when conflict happened, how was it resolved at this stage? Also, was any sort of software management system used for the review process or in the study?
Reviewer 2 Report
Comments and Suggestions for Authors
Dear authors,
Thank you very much for submission of your paper. It is very well-written and important information that can warrant future education and intervention strategies for collegiate athletes. Please see comments below:
References: Not in correct style for the journal. Please see instructions to authors for
referencing guidelines.
Abstract: One minor edit:
- Line 23: change to “interventions”
Introduction:
- Delete: “ National surveys reveal significant FI among college athletes;”
- All student-athletes? Or specifically varsity or club?
Methods:
- Line 96: capitalize F in figure.
- Section 2.3 – Figure needs to be labeled; and I recommend improving the formatting of the arrows within the figure
Results:
- Line 111: change to “…88 participants per study”
- Move Table 1 up right after section 3.1
- Be sure to define spell out NCAA before you abbreviate it in text
- Spell out ROTC because you abbreviate it
- I would like to know which articles within the table are published and which unpublished (theses/dissertations/etc)
Table 1: personally, I don’t love bullet points when they are centered. If you choose to use bullet points, please left-justify all of the text. Or remove the bullets and keep entered
Discussion:
- I’d be interested in reading about differences across division. I’ve worked with student-athletes at a DIII school and a DI school and they are drastically different in their resources. DIII the one cafeteria closed early, before practice would even end, where the DI school had athlete-only cafeterias which chef’s to make them whatever they wanted. Plus more NIL money coming in. Can you add some divisional differences within these sections of your discussion? This would shef light on more specific future targeted intervention strategies.
